# Factors Related to Mpox-Vaccine Uptake among Men Who Have Sex with Men in Taiwan: Roles of Information Sources and Emotional Problems

**DOI:** 10.3390/vaccines12030332

**Published:** 2024-03-20

**Authors:** Mei-Feng Huang, Yu-Ping Chang, Chien-Wen Lin, Cheng-Fang Yen

**Affiliations:** 1Department of Psychiatry, Kaohsiung Medical University Hospital, Kaohsiung Medical University, Kaohsiung 80708, Taiwan; 2Department of Psychiatry, School of Medicine, College of Medicine, Kaohsiung Medical University, Kaohsiung 80708, Taiwan; 3School of Nursing, University at Buffalo, The State University of New York, New York, NY 14214, USA; 4College of Professional Studies, National Pingtung University of Science and Technology, Pingtung 912301, Taiwan

**Keywords:** mpox, vaccine, men who have sex with men

## Abstract

An mpox outbreak occurred suddenly and rapidly spread worldwide in 2022. Research has demonstrated a link between the sexual behavior of men who have sex with men (MSM) and the contraction of mpox. This study assessed the factors related to mpox-vaccine uptake among MSM in Taiwan, focusing on the roles of information sources and emotional problems. In total, 389 MSM participated in an online survey. Data on the participants’ vaccination statuses; anxiety symptoms, which were assessed using the State–Trait Anxiety Inventory; depressive symptoms, which were assessed using the Center for Epidemiologic Studies Depression Scale; and risk perceptions of contracting mpox were collected. Factors related to mpox-vaccine uptake were examined using a multivariable logistic regression model. The results revealed that MSM who were older (*p* < 0.001), perceived a higher risk of contracting mpox (*p* = 0.040), and received mpox information from health-care providers (*p* < 0.001) were more likely to receive mpox vaccination, whereas MSM who reported a greater severity of depression (*p* = 0.017) were less likely to receive mpox vaccination. However, age did not moderate the associations of perceiving a higher risk of contracting mpox, receiving mpox information from health-care providers, and depression with having an mpox vaccination. Health-care providers should consider these factors when developing intervention programs for enhancing mpox-vaccine uptake among MSM.

## 1. Introduction

The mpox virus was first detected among research monkeys in 1958. The earliest case of mpox infection in humans was in a baby in the Congo in 1970, and since then, cases have been reported in Central and West Africa [1]. An mpox outbreak occurred suddenly and rapidly spread worldwide in May 2022 [1]. From 1 January 2022 to 30 November 2023, a total of 92,783 laboratory-confirmed mpox cases (including 171 deaths) from 116 countries were reported to the World Health Organization (WHO) [1]. Common early symptoms of mpox infection include pain, fever, fatigue, and lymphadenectasis, with significant inguinal lymphadenectasis often observed. Following the latent period, individuals infected with mpox begin to exhibit atypical symptoms, including fever and chills, headache, muscle pain, and lymphadenectasis. After the fever and lymphadenectasis, rashes begin to appear on the head and face and gradually spread throughout the body. The rash evolves from papules to vesicles and pustules, ultimately forming crusts that heal, leaving behind scars [2]. However, the clinical presentation of mpox cases with this outbreak has been atypical compared with previously documented reports. Many cases in affected areas are not presenting with the classically described clinical picture for mpox, and few lesions, or no lesions but anal pain and bleeding, genital and perineal/perianal alone lesions, lesions at different stages of development, and absence of prodromal period or constitutional symptoms appearing after lesions are observed. The number of suspected cases of mpox infections that had not been confirmed by laboratory testing was much higher. For example, from 1 January to 12 November 2023, a total of 12,569 suspected mpox cases, including 581 suspected mpox deaths, were reported in the Democratic Republic of the Congo [3]. Moreover, the age group of infected individuals had changed from predominantly children to young adults [4], especially men who have sex with men (MSM) and individuals with human immunodeficiency virus (HIV) infection [5,6,7]. The aforementioned statistics indicate that mpox is an infectious disease that cannot be ignored in today’s world.

According to the WHO, receiving vaccination and avoiding physical contact with an individual having mpox can prevent infection in individuals at risk [1]. In 2019, the U.S. Food and Drug Administration approved a new vaccine (JYNNEOS) containing live attenuated vaccinia virus for the prevention of smallpox and mpox in high-risk individuals aged more than 18 years [6]. JYNNEOS is an attenuated, live, non-replicating smallpox and monkeypox vaccine that elicits humoral and cellular immune responses to orthopoxviruses. Research found that the adjusted vaccine effectiveness was 75% for one dose and 86% for two doses of JYNNEOS vaccine, indicating substantial protection against mpox, irrespective of route of administration or immunocompromised status [8]. Therefore, persons at high risk for mpox exposure should be vaccinated with the recommended two-dose JYNNEOS series.

As of November 2023, there have been 355 confirmed cases of mpox in Taiwan (where the present study was carried out), including one death [9]. Due to the mpox outbreak, Taiwan has imported the mpox vaccine and provides it at public expense to those who meet the eligibility criteria, such as (i) being over 18 years of age who have received pre-exposure prophylaxis and post-exposure prophylaxis against HIV, (ii) having engaged in risky sexual behaviors in the past year, (iii) having had sexually transmitted diseases through past sexual contacts, and (iv) being a health-care professional who has direct contact with those who have mpox [9]. Taiwan has also set up the mpox vaccination clinics in several hospitals to minimize the difficulty of the public in obtaining the vaccination.

A meta-analysis of 29 studies involving 52,658 participants demonstrated that the pooled prevalence of the intention to vaccinate against mpox was 61% around the world [10]. However, people’s willingness to receive mpox vaccination does not guarantee that they will get vaccinated [11]. Research of the factors related to mpox-vaccine uptake among MSM can provide evidence for developing intervention strategies to enhance mpox-vaccine uptake. According to protection motivation theory (PMT), both threat appraisal and coping appraisal determine individuals’ intention to adopt protective behaviors for reducing the risk of contracting mpox [12,13]. Threat appraisal depends on the perceived severity of the health threat caused by mpox and the perceived vulnerability to mpox. It is well known that the individuals who perceive greater severities and vulnerabilities of contracting infectious diseases are more likely to adopt protective behaviors against infectious diseases [14]. Coping appraisal depends on the perceived response efficacy (i.e., the evaluation of whether self-protective behaviors and receiving a vaccination are effective in alleviating the threat of mpox). Research found that perceived knowledge of respiratory diseases is associated with coping appraisal, and coping appraisal is associated with adaptive responses against respiratory diseases [15]. It is reasonable to hypothesize that the factors influencing individuals’ threat appraisal and coping appraisal toward mpox and mpox vaccines relate to uptake of mpox vaccination.

Several studies have examined the factors related to the intention to vaccinate against mpox. Studies in the general population have revealed that sociodemographic characteristics (such as male sex, younger age, nonheterosexual orientation, higher education level, and urban residence); cognitive appraisal (such as higher perceived susceptibility to and severity of mpox, increased knowledge of mpox, lower conspiracy beliefs about mpox, greater emotional fear of mpox, and higher perceived potential of the occurrence of a pandemic); confidence in vaccination; and collective responsibility are significantly associated with a higher intention to vaccinate against mpox [16,17,18,19,20,21]. Studies involving MSM have revealed that those ones who are gay, spend more time with other gay or bisexual men, have good knowledge of mpox, have higher perceived susceptibility to mpox, have no steady relationship, and have low constraints on vaccine access are more likely to receive mpox vaccination [11,22,23]. The results of studies in the general population and MSM have confirmed the proposition of PMT that perceived susceptibility to and severity of mpox and perceived benefits and barriers of vaccination contribute to the intention to vaccinate and the uptake of vaccines against mpox [12,13].

Several issues regarding the factors related to mpox-vaccine uptake among MSM warrant research. First, the role of sources of information on mpox in the uptake of vaccines has not been examined. According to PMT, the knowledge of mpox influences threat and coping appraisals among MSM [12,13]. Research in the general population has revealed that health-care professionals and officials are the most trusted sources of information on the mpox outbreak, followed by well-known doctors and researchers with a large online following [20]. A study of the coronavirus disease 2019 (COVID-19) pandemic also found that receiving information concerning COVID-19 vaccination from medical personnel was associated with greater self-efficacy, response efficacy, and knowledge, whereas receiving information concerning COVID-19 vaccination from coworkers/colleagues was associated with less response efficacy and knowledge [14]. Social media is an important source of health information in the modern world. People seek health information in social media; they ranged from online discussions on specific diseases to public health concerns [24]. Benefits of health seeking on social media, in addition to filling a need for health information, include the social and emotional support health consumers gain from peer-to-peer interactions [24]. However, health misinformation is prevalent on social media [20]. Both vaccines and disease pandemics are main issues of health misinformation on social media; people can easily become misinformed and delay the timing of adopting self-protective behaviors, including receiving a vaccination [25]. The associations between receiving information on mpox from social media and from health-care providers and mpox-vaccine uptake need to be studied further. Second, a study revealed that lesbian, gay, bisexual, transgender, and queer individuals with severe or moderate depression and anxiety were less likely to receive COVID-19 vaccination [26]. According to PMT, emotional problems may reduce the self-efficacy of individuals in adopting coping strategies and overcoming the difficulty of receiving vaccination [12,13]. Emotional problems may also reduce individuals’ motivation to search for information and interact with others; thus, they may lack real-time information regarding the severity of infectious diseases and vaccines. However, whether depression and anxiety are significantly related to mpox-vaccine uptake has not been examined.

As mentioned in the earlier text, this study assessed the factors related to mpox-vaccine uptake among MSM in Taiwan, focusing on the roles of information sources and emotional problems. We hypothesized that sociodemographic characteristics (such as sex, age, sexual orientation, and education level); perceived risk of contracting mpox; emotional problems (such as depression and anxiety); and mpox information sources (such as social media and health-care providers) are significantly associated with mpox-vaccine uptake among MSM.

## 2. Methods

### 2.1. Participants

In this study, participants were recruited by posting a link to the research survey website on Facebook, LINE, and the Professional Technology Temple Bulletin Board System from 1 November to 31 December 2023. Taiwanese men who were ≥20 years old and who had engaged in sexual intercourse with men over the past year were included. Men who have sex with men who were willing to participate in this study could press the “agree to participate” button in the advertisement and could complete the survey after providing their informed consent. Individuals who were unwilling to participate in the study could press the “refuse to participate” button and leave. A total of 389 MSM voluntarily participated and completed the survey. This study was approved by the Institutional Review Board (IRB) of Kaohsiung Medical University Hospital (KMUHIRB-EXEMPT(I)-20230008).

### 2.2. Measures

#### 2.2.1. Vaccination Status

To assess their current vaccination status, the participants were asked “Have you been vaccinated against mpox?” They responded with either “no” or “yes”.

#### 2.2.2. Emotional Problems

The Chinese version of the 10-item State–Trait Anxiety Inventory—State version was used to assess the participants’ anxiety symptoms in the past month [27]. Each item is rated on a four-point scale, with a higher total score indicating a higher level of anxiety. In this study, Cronbach’s α was 0.88. The Chinese version of the 10-item Center for Epidemiologic Studies Depression Scale was used to assess the participants’ depressive symptoms in the past month [28]. Each item is rated on a four-point scale, with a higher total score indicating higher levels of depressive symptoms. In this study, Cronbach’s α was 0.90.

#### 2.2.3. Risk Perception of Contracting Mpox

To assess the risk perception of contracting mpox, the participants were asked a single question: “What do you think are your chances of contracting mpox in the next month?” This item was rated on a five-point scale, ranging from 1 (“not at all”) to 5 (“extremely high”).

#### 2.2.4. Demographics

The demographic characteristics of participants were also collected, including their age (in years), educational level (senior high school or below versus college or above), and sexual orientation (gay versus bisexual).

### 2.3. Data Analysis

Descriptive statistics (means, standard deviations, frequencies, and percentages) were used to summarize the participants’ characteristics. The associations between independent variables (such as age, sexual orientation, education level, depression, anxiety, perceived risk of contracting mpox, and receiving mpox information from social media or health-care providers) and dependent variables (such as mpox-vaccine uptake) were examined using bivariable and multivariable logistic regression models. The moderating effects of demographics and sexual orientation on the associations of information sources and emotional problems were also examined. Multicollinearity between the independent variables should be checked by calculating the variance inflation factor (VIF). A *p* value lower than 0.05 was considered statistically significant. All data analyses were performed using SPSS 24.0 (SPSS Inc., Chicago, IL, USA).

## 3. Results

This study included 389 participants, with a mean age of 33.72 years (standard deviation = 6.42). Moreover, most participants were gay (n = 343; 88.2%). Almost all participants (91.0%) had an education level of college or above. The participants’ scores of anxiety and depression and their perceived risk of contracting mpox in the next month are provided in Table 1. More than four-fifths (88.9%) of the participants had ever received mpox information from social media, and more than half (55.8%) of the participants had ever received mpox information from health-care providers. More than (59.9%) of the participants had received mpox vaccination.

Table 2 presents the results of the bivariable logistic regression analysis examining the associations of the factors with mpox-vaccine uptake. The participants who were older (*p* < 0.001), perceived a greater risk of contracting mpox (*p* = 0.042), and received mpox information from health-care providers (*p* < 0.001) were more likely to receive mpox vaccination. Sexual orientation, education level, anxiety, depression, and receiving mpox information from social media were not significantly associated with mpox-vaccine uptake (*p* > 0.05).

Table 3 presents the results of multivariable logistic regression analysis examining the associations of the factors with mpox-vaccine uptake. The value of VIF was 23.962, indicating no collinearity, as determined on the basis of the suggestions of Senaviratna and Cooray [29]. The results of Model I indicated that, after adjusting for other factors, participants who perceived a greater risk of contracting mpox (*p* = 0.040) and had received mpox information from health-care providers (*p* < 0.001) were more likely to receive mpox vaccination, whereas those who reported a greater severity of depression were less likely to receive mpox vaccination (*p* = 0.017). Anxiety, receiving mpox information from social media, sexual orientation, and education level were not significantly associated with mpox-vaccine uptake (*p* > 0.05). The interactions of age with depression, perceived risk of contracting mpox, and receiving mpox information from health-care providers were further entered into a multivariable logistic regression analysis (Model II) to examine the moderating effects of age. However, the interactions were not significantly associated with receiving an mpox vaccination (*p* > 0.05), indicating that age did not moderate the associations of depression, perceived risk of contracting mpox, and receiving mpox information from health-care providers with receiving an mpox vaccination. We also transformed age into five-year increments of age and its association with receiving an mpox vaccination using multivariable logistic regression analysis (Appendix A). The result indicated that five-year increments of age were still significantly associated with receiving an mpox vaccination (*p* < 0.001).

## 4. Discussion

This study revealed that MSM who perceived a greater risk of contracting mpox were more likely to adopt protective behaviors against mpox, such as receiving mpox vaccination. Moreover, MSM who were older, received mpox information from health-care providers, and had lower levels of depression were more likely to receive mpox vaccination.

Positive associations were noted between the perceived risk of contracting mpox and mpox-vaccine uptake among MSM. Previous studies have also found that MSM who have higher perceived susceptibility to mpox are more likely to receive mpox vaccination [11,23]. According to PMT, the perceived risk of contracting an infectious disease denotes a threat appraisal that increases individuals’ intention to adopt protective behaviors [12,13]. Given the increased prevalence of HIV among this population and the recent COVID-19 pandemic, the mpox outbreak in 2023 may have induced fear among MSM. In particular, pictures of skin lesions in infected individuals were spread through media, making MSM experience fear regarding the risk of infection. Since mpox is most often transmitted through sexual contacts, the message of contracting mpox is especially likely to spread rapidly among the MSM community, resulting in a deep sense of threat for MSM. Although the perceived risk of contracting mpox increases the intention for this group of people to adopt protective behaviors, overanxiety about the possibility of contracting mpox may compromise their ability to respond to the outbreak appropriately. Ensuring that MSM perceive the risk of contracting mpox without being overly alarmed is an important task for medical professionals when promoting mpox knowledge.

This study revealed that MSM who received mpox information from health-care providers were more likely to receive mpox vaccination. According to PMT, knowledge is essential to coping appraisal and development of coping strategies [12,13]. Men who have sex with men who received mpox information from health-care providers may have obtained immediate and correct knowledge of mpox, which may have increased their intention to receive mpox vaccination. A previous study also found that good knowledge of mpox is associated with receiving an mpox vaccination [22]. Receiving mpox information from health-care providers is also indicative of access to the mpox vaccine. A previous study also found that MSM who have low constraints on vaccine access are more likely to receive mpox vaccination [11]. Alternatively, MSM may have received mpox information upon visiting health-care providers for vaccination. Although the cross-study design limited the possibility of determining the causal relationship between receiving mpox information from health-care providers and mpox-vaccine uptake, the findings of this study indicate the important role of health-care providers in disseminating acute information on mpox and increasing mpox-vaccine uptake among MSM. This study did not reveal a significant association between receiving mpox information from social media and mpox-vaccine uptake among MSM. Men who have sex with men interact with a high percentage of other MSM on social media, resulting in a more consistent focus on mpox and reducing the chance of misinformation circulating on social media. In this study, a high proportion (88.9%) of MSM received mpox information from social media. Given that health misinformation is prevalent on social media, mental health professionals should evaluate what information MSM have received from social media and the accuracy of the received information regarding mpox and mpox vaccines [25]. Alternatively, the pervasive use of social media indicates that social media is a suitable channel for promoting appropriate knowledge on mpox and vaccines.

This study also revealed that MSM who reported a greater severity of depression were less likely to receive mpox vaccination. Men who have sex with men who experience depression may reduce their interactions with peers and their access to sources of information on mpox. According to PMT, no immediate or correct knowledge of mpox and vaccines may result in delayed mpox vaccination [12,13]. Men who have sex with men who experience depression may also lack the confidence and executive force to receive an mpox vaccine. Not receiving an mpox vaccine may also be a sign of giving up the will to live. In this study, a positive association was noted between age and mpox-vaccine uptake among MSM. Young adults were predominantly at risk of contracting mpox in the 2023 outbreak [4]. Our study results indicated that among MSM, young adults and individuals with depression should be the target of interventions for enhancing mpox-vaccine uptake.

The present study did not find a significant association between sexual orientation and having an mpox vaccination. The result was in contrast to the result of a previous study on MSM [11]. Moreover, previous studies have found that MSM who spend time with other gay or bisexual men and have steady relationships are more likely to have an mpox vaccination [11,23]. It is possible that MSM who spend more time with other gay or bisexual men can perceive the norm of being vaccinated against mpox and obtain accurate knowledge about the vaccination and thus have a higher motivation to receive an mpox vaccination. Having steady relationships may increase MSM’s motivation to receive an mpox vaccination and protect their partners. However, the present study did not examine participants’ time spent on interacting with other gay or bisexual men or their relationship status. Further study is needed to examine them. The present study had similar findings, such as the associations between perceived susceptibility to and severity of mpox and the intention to vaccinate against mpox, compared to previous studies in the general population [19,20]. However, the present study found that older MSM were more likely to receive an mpox vaccination, and the result was contrary to that of two previous studies, indicating the difference in the factors related to the intention to vaccinate against mpox between the general and MSM populations [17,18].

The present study is the first one to examine the roles of sources of information and emotional problems on mpox in the uptake of vaccines in MSM. The results of this study provide evidence for developing intervention programs to enhance MSM’s intention to receive an mpox vaccination. However, this study has several limitations. The study participants were recruited using an online advertisement and through convenience sampling. Therefore, sampling bias may have occurred, and the sample may not necessarily be representative of MSM. The sample included only individuals who had access to the Internet; this may have led to biasing the sample toward the inclusion of younger individuals. Most of the participants had an education level of college or above. Whether the results of this study can be generalized to the MSM population with an education level of high school or below warrants further study. Future studies with larger and more representative samples are warranted. Moreover, all data were self-reported by the participants. Therefore, the researchers could not fully control for single-rater bias. The participants may also have provided socially desirable responses instead of choosing responses reflective of their true feelings. Potential information bias should be examined in further studies.

## 5. Conclusions

This study demonstrated that MSM who are older, perceive a higher risk of contracting mpox, and receive mpox information from health-care providers are more likely to receive mpox vaccination, whereas MSM who reported a greater severity of depression were less likely to receive mpox vaccination. As mpox is still spreading, health-care providers should integrate the related factors into intervention programs to enhance mpox-vaccine uptake among MSM. Health-care providers could also increase MSM’s knowledge regarding the risk of contracting mpox and improve MSM’s depression to enhance their motivation to vaccinate against mpox.

## Figures and Tables

**Table 1 vaccines-12-00332-t001:** Participant demographic characteristics, perceived risk of contracting mpox, emotional problems, information sources, and vaccination status (N = 389).

	Mean (SD) or *n* (%)
Age (in years)	33.72 (6.42)
Sexual orientation	
Gay	343 (88.2)
Bisexual	46 (11.8)
Educational level	
≤senior high	35 (9.0)
≥college	354 (91.0)
Anxiety	20.56 (7.17)
Depression	9.00 (5.35)
Perceived risk of contracting mpox	0.98 (0.82)
Receiving information regarding mpox from social media	346 (88.9)
Receiving information regarding mpox from health-care providers	217 (55.8)
Vaccination status	
Not vaccinated	156 (40.1)
Vaccinated	233 (59.9)

**Table 2 vaccines-12-00332-t002:** Factors related to monkeypox (mpox)-vaccine uptake: bivariable logistic regression models (N = 389).

	Receiving an Mpox Vaccination	
	OR (95% CI)	*p*
Age	1.062 (1.026–1.099)	0.001
Sexual orientation ^a^	0.854 (0.459–1.591)	0.619
Education level ^b^	0.761 (0.367–1.578)	0.463
Anxiety	0.989 (0.962–1.018)	0.452
Depression	0.965 (0.929–1.003)	0.067
Perceived risk of contracting mpox	1.305 (1.010–1.688)	0.042
Receiving mpox information from social media	1.208 (0.637–2.289)	0.563
Receiving mpox information from health-care providers	4.398 (2.853–6.780)	<0.001

CI: confidence interval; OR: odds ratio; ^a^ gay as reference; ^b^ senior high school or below as reference.

**Table 3 vaccines-12-00332-t003:** Factors related to monkeypox (mpox)-vaccine uptake: multivariable logistic regression models (N = 389).

	Receiving an Mpox Vaccination			
	Model I		Model II	
	OR (95% CI)	*p*	OR (95% CI)	*p*
Age	1.072 (1.033–1.113)	<0.001	1.145 (1.048–1.251)	0.003
Sexual orientation ^a^	0.919 (0.457–1.848)	0.813	0.881 (0.434–1.787)	0.725
Education level ^b^	0.853 (0.376–1.933)	0.703	0.857 (0.375–1.955)	0.713
Anxiety	1.018 (0.975–1.063)	0.406	1.021 (0.977–1.066)	0.356
Depression	0.931 (0.879–0.987)	0.017	1.163 (0.911–1.485)	0.227
Perceived risk of contracting mpox	1.349 (1.014–1.794)	0.040	2.444 (0.543–11.007)	0.245
Receiving mpox information from social media	0.679 (0.331–1.394)	0.292	0.670 (0.327–1.375)	0.275
Receiving mpox information from health-care providers	5.263 (3.280–8.444)	<0.001	1.432 (0.107–19.209)	0.786
Age × Depression			0.993 (0.986–1.000)	0.066
Age × Perceived risk of contracting mpox			0.982 (0.938–1.027)	0.422
Age × Receiving mpox information from health-care providers			1.042 (0.963–1.126)	0.307

CI: confidence interval; OR: odds ratio; ^a^ gay as reference; ^b^ senior high school or below as reference.

## Data Availability

The data are available upon reasonable request to the corresponding authors.

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
