# Peer review of "Factors Related to Mpox-Vaccine Uptake among Men Who Have Sex with Men in Taiwan: Roles of Information Sources and Emotional Problems"

_vaccines, 2024, doi:10.3390/vaccines12030332_

Round 1

Reviewer 1 Report

Comments and Suggestions for Authors

This study assessed the factors related to mpox vaccine uptake among MSM in Taiwan, focusing on the roles of information sources and emotional problems, and revealed that MSM who were older (p < 0.001), perceived a higher risk of contracting mpox (p = 0.040), and received mpox information from health-care providers (p < 0.001) were more likely to receive mpox vaccination, whereas MSM who reported a greater severity of depression (p = 0.017) were less likely to receive mpox vaccination. The finding is of clinical significance; however, some issues maintained to be discussed.

1. Main issue: Factors such as information sources and emotional problems may interacted with each other; thus, multicollinearity between the independent variables should be checked by calculating the variance inflation factor (VIF).

2 Minor issue (lines 151-157): Related p values are suggested to be added up to the manuscript.

Author Response

We appreciated your valuable comments. As discussed below, we have revised our manuscript with underlines based on your suggestions. Please let us know if we need to provide anything else regarding this revision.

Comment 1

Main issue: Factors such as information sources and emotional problems may interacted with each other; thus, multicollinearity between the independent variables should be checked by calculating the variance inflation factor (VIF).

Response

Thank you for your comment. We added the result of examining multicollinearity by calculating the VIF into the revised manuscript.

Methods

Multicollinearity between the independent variables should be checked by calculating the variance inflation factor (VIF).” Please refer to line 183-184.

Results

The value of VIF was 23.962, indicating no collinearity, as determined on the basis of the suggestions of Senaviratna and Cooray [29].” Please refer to line 209-211.

Comment 2

Minor issue (lines 151-157): Related p values are suggested to be added up to the manuscript.

Response

Thank you for your comment. We added p values here. Please refer to line 199-204.

“The participants who were older (p < 0.001), perceived a greater risk of contracting mpox (p = 0.042), and receiving mpox information from healthcare providers (p < 0.001) were more likely to receive mpox vaccination. Sexual orientation, education level, anxiety, depression, and receiving mpox information from social media were not significantly associated with mpox vaccine uptake (p > 0.05).”

Reviewer 2 Report

Comments and Suggestions for Authors

This is a nice and concise report describing factors that are associated with mpox vaccine uptake among MSM in Taiwan.

The methodology and analysis is sound, and the results and conclusions are clearly presented and appropriate. Limitations of the study are adequately acknowledge.

The main shortcoming of the study relates to the recruitment of the study participants, who were exclusively recruited on a voluntary basis through online questionnaires. This will almost certainly have introduced an undesirable bias in the survey. However, this shortcoming is acknowledged by the authors.

A minor shortcoming is that authors fail to compare their results with any similar studies in other countries/locations. This would be relevant to understand whether Taiwan represents a unique situation or similar to other locations  

Author Response

We appreciated your valuable comments. As discussed below, we have revised our manuscript with underlines based on your suggestions. Please let us know if we need to provide anything else regarding this revision.

Comment 1

The main shortcoming of the study relates to the recruitment of the study participants, who were exclusively recruited on a voluntary basis through online questionnaires. This will almost certainly have introduced an undesirable bias in the survey. However, this shortcoming is acknowledged by the authors.

Response

Thank you for your comment. We agree that the recruitment method is one of the limitations of this study. In addition to the original description, we added the recruitment of a group of MSM with a high educational level as the limitation of this study. Please refer to line 311-313.

“The study participants were recruited using an online advertisement and through convenience sampling. Therefore, sampling bias may have occurred, and the sample may not necessarily be representative of MSM. The sample included only individuals who had access to the Internet; this may have led to biased the sample toward the inclusion of younger individuals. Most of the participants had an education level of college or above. Whether the results of this study can be generalized to the MSM population with an education level of high school or below warrants further study.

Comment 2

A minor shortcoming is that authors fail to compare their results with any similar studies in other countries/locations. This would be relevant to understand whether Taiwan represents a unique situation or similar to other locations.

Response

Thank you for your comment. We added discussions to compare the results of the present study with those of previous studies in Discussion.

  • Previous studies have also found that MSM who have higher perceived susceptibility to mpox are more likely to receive mpox vaccination [11,23].” Please refer to line 235-237.
  • A previous study also found that good knowledge of mpox is associated with receiving a mpox vaccination [22].” Please refer to line 255-256.
  • A previous study also found that MSM who have low constraints on vaccine access are more likely to receive mpox vaccination [11].” Please refer to line 257-259.
  • In contrast to the result of a previous study on MSM [11], the present study did not find a significant association between sexual orientation and having a mpox vaccination. Moreover, previous studies have found that MSM who spend time with other gay or bisexual men and have steady relationship are more likely to have a mpox vaccination [11,23]. It is possible that MSM who spend more time with other gay or bisexual men can perceive the norm of being vaccinated against mpox and obtain accurate knowledge about the vaccination and thus have a higher motivation to receive a mpox vaccination. Have steady relationship may increase MSM’s motivation to receive a mpox vaccination and protect their partners. However, the present study did not examine participants’ time spent on interacting with other bay or bisexual men or the relationship status. Further study is needed to examine them. Compared with previous studies in the general population [19,20], the present study had similar findings such as the associations between perceived susceptibility to and severity of mpox and the intention to vaccinate against mpox. However, the present study found that older MSM were more likely to receive a mpox vaccination, and the result was contrary to that of two previous studies [17,18], indicating the difference in the factors related to the intention to vaccinate against mpox between the general and MSM populations.” Please refer to line 286-302.

Reviewer 3 Report

Comments and Suggestions for Authors

This study assessed the factors related to mpox vaccine uptake among MSM in Taiwan. The study focused on the roles of information sources and anxiety and depression.

The study provides a clear statement of the problem, however there is some comments for authors:

Abstract

It is written that MSM are at a higher risk of contracting mpox than other cohorts. Other cohorts need clarification.

Introduction

Place all reference numbers before the punctuation.

Results

Table 1. The table title it must be more informative about the participant characteristics.

Discussion

About the information source the authors underline (lines 83 and 84) “Social media is an important source of health information in the modern world: However misinformation is prevalent on social media”. However (lines 191 to 193) it’s written “In this study, a high proportion (88.9%) of MSM received mpox information from social media, indicating that social media is a suitable channel for promoting appropriate knowledge on mpox and vaccines”. The two statements don’t match one with the other.

Conclusion

The statement (lines 216 to 218) “This study demonstrated the age, depression, perceived risk of contracting mpox, and receiving mpox information from healthcare provides are related to mpox vaccine uptake among MSM”, it’s not clear and looks contradictory. Before (lines 194 and 195) is written “This study also revealed that MSM who reported a greater severity of depression were less likely to receive mpox vaccination”. Concerning age, which age is related to mpox vaccine uptake among MSM?

Author Response

We appreciated your valuable comments. As discussed below, we have revised our manuscript with underlines based on your suggestions. Please let us know if we need to provide anything else regarding this revision.

Comment 1

Abstract

It is written that MSM are at a higher risk of contracting mpox than other cohorts. Other cohorts need clarification.

Response

Thank you for your comment. We changed this sentence as below. Please refer to line 15-16.

Research has demonstrated a link between the sexual behavior of men who have sex with men (MSM) and the contraction of mpox.

Comment 2 

Introduction

Place all reference numbers before the punctuation.

Response

Thank you for your reminding. We rechecked the text and place all reference numbers before the punctuation.

Comment 3

Results

Table 1. The table title it must be more informative about the participant characteristics.

Response

Thank you for your comment. We changed the table title and made it be more informative about the participant characteristics. Please refer to line 196-197.

Table 1. Participant demographic characteristics, perceived risk of contracting mpox, emotional problems, information sources, and vaccination status (N = 389).

Comment 4

Discussion

About the information source the authors underline (lines 83 and 84) “Social media is an important source of health information in the modern world: However misinformation is prevalent on social media”. However (lines 191 to 193) it’s written “In this study, a high proportion (88.9%) of MSM received mpox information from social media, indicating that social media is a suitable channel for promoting appropriate knowledge on mpox and vaccines”. The two statements don’t match one with the other.

Response

Thank you for your comment. We revised the sentences as below to match one with the other. Please refer to line 268-274.

In this study, a high proportion (88.9%) of MSM received mpox information from social media. Given that health misinformation is prevalent on social media [25], mental health professionals should evaluate what information MSM have received from social media and the accuracy of the received information regarding mpox and mpox vaccines. Alternatively, the pervasive use of social media indicates that social media is a suitable channel for promoting appropriate knowledge on mpox and vaccines.

Comment 5

Conclusion

The statement (lines 216 to 218) “This study demonstrated the age, depression, perceived risk of contracting mpox, and receiving mpox information from healthcare provides are related to mpox vaccine uptake among MSM”, it’s not clear and looks contradictory. Before (lines 194 and 195) is written “This study also revealed that MSM who reported a greater severity of depression were less likely to receive mpox vaccination”. Concerning age, which age is related to mpox vaccine uptake among MSM?

Response

Thank you for your comment. We revised the sentence to make its meaning clearer. Please refer to line 319-322.

This study demonstrated that MSM who are older, perceive a higher risk of contracting mpox, and receive mpox information from health-care providers are more likely to receive mpox vaccination, whereas MSM who reported a greater severity of depression were less likely to receive mpox vaccination.

Reviewer 4 Report

Comments and Suggestions for Authors

This is an interesting study on the adoption of vaccine. The study is well conducted, given the limitations of conducting an online survey. The study relies on an acceptable theoretical model and the findings are interesting. Some suggestions to the authors

1. Education seems not to be a variable. Most of the respondents reported high education. Then it is not surprising that the variable is non significant because there is no enough variation. I suggest to discuss in the study limitations this limitation as well.

2. The author/s note that they rely on a theoretical model however do not discuss the implications of the findings for the theoretical model. I suggest to do this.

3. The study relied on msm. I wonder if the results will differ if the study was conducted for the whole population. Can the authors discuss this as well?

I hope the comments are helpful.

Author Response

We appreciated your valuable comments. As discussed below, we have revised our manuscript with underlines based on your suggestions. Please let us know if we need to provide anything else regarding this revision.

Comment 1

Education seems not to be a variable. Most of the respondents reported high education. Then it is not surprising that the variable is non significant because there is no enough variation. I suggest to discuss in the study limitations this limitation as well.

Response

Thank you for your comment. We listed it as one of the limitations in the revised manuscript. Please refer to line 307-310.

Most of the participants had an education level of college or above. Whether the results of this study can be generalized to the MSM population with an education level of high school or below warrants further study.

Comment 2

The author/s note that they rely on a theoretical model however do not discuss the implications of the findings for the theoretical model. I suggest to do this.

Response

Thank you for your comment. We added discussion regarding to the implications of the findings for the theoretical model.

According to PMT [12,13], the perceived risk of contracting an infectious disease denotes a threat appraisal that increases individuals’ intention to adopt protective behaviors. Given the increased prevalence of HIV among this population and the recent COVID-19 pandemic, the mpox outbreak in 2023 may have induced fear among MSM. In particular, pictures of skin lesions in infected individuals were spread through media, making MSM experience fear regarding the risk of infection. Since mpox is most often transmitted through sexual contacts, the message of contracting mpox is especially likely to spread rapidly among the MSM community, resulting in a deep sense of threatening for MSM.” Please refer to line 237-244.

According to PMT [12,13], knowledge is essential to coping appraisal and development of coping strategies. MSM who received mpox information from health-care providers may have obtained immediate and correct knowledge of mpox, which may have increased their intention to receive mpox vaccination. A previous study also found that good knowledge of mpox is associated with receiving a mpox vaccination [22]. Receiving mpox information from health-care providers is also indicative of access to the mpox vaccine. A previous study also found that MSM who have low constraints on vaccine access are more likely to receive mpox vaccination [11].” Please refer to line 251-259.

This study also revealed that MSM who reported a greater severity of depression were less likely to receive mpox vaccination. MSM who experience depression may reduce their interactions with peers and their access to sources of information on mpox. According to PMT [12,13], no immediate or correct knowledge of mpox and vaccines may result in delayed mpox vaccination.” Please refer to line 275-279.

Comment 3

The study relied on msm. I wonder if the results will differ if the study was conducted for the whole population. Can the authors discuss this as well?

Response

Thank you for your comment. We added a new paragraph in Discussion for comparing the results of present study with those of previous studies involving MSM and the general population. Please refer to line 286-302.

In contrast to the result of a previous study on MSM [11], the present study did not find a significant association between sexual orientation and having a mpox vaccination. Moreover, previous studies have found that MSM who spend time with other gay or bisexual men and have steady relationship are more likely to have a mpox vaccination [11,23]. It is possible that MSM who spend more time with other gay or bisexual men can perceive the norm of being vaccinated against mpox and obtain accurate knowledge about the vaccination and thus have a higher motivation to receive a mpox vaccination. Have steady relationship may increase MSM’s motivation to receive a mpox vaccination and protect their partners. However, the present study did not examine participants’ time spent on interacting with other bay or bisexual men or the relationship status. Further study is needed to examine them. Compared with previous studies in the general population [19,20], the present study had similar findings such as the associations between perceived susceptibility to and severity of mpox and the intention to vaccinate against mpox. However, the present study found that older MSM were more likely to receive a mpox vaccination, and the result was contrary to that of two previous studies [17,18], indicating the difference in the factors related to the intention to vaccinate against mpox between the general and MSM populations.”

Reviewer 5 Report

Comments and Suggestions for Authors

This is a well written paper and I only have some suggestions/edits to recommend to the authors:

1. On line 138 the authors wrote "...examined using multivariate logistic regression models". Is is multivariate or multivariable? There is a major difference between the two methodologies.

2. In the methods section the authors have not specified whether they have tested for effect modification. To this end, they should examine one of the independent variables used in the model as a potential exposure of interest and then identify the corresponding confounders of interest. This is a very important step in the analysis, since it can provide further information on the association under investigation.

3. In the multivariable logistic regression model, I suggest testing also the 5-year increment of age, since it can reveal essential information otherwise not captured in the model.

4. At the end of the discussion section the authors talk about the study limitations but do not reflect on the study strengths. What are this study's strengths? This information needs to be included as well.

Comments on the Quality of English Language

There are only minor English language edits needed for this paper.

Author Response

We appreciated your valuable comments. As discussed below, we have revised our manuscript with underlines based on your suggestions. Please let us know if we need to provide anything else regarding this revision.

Comment 1

On line 138 the authors wrote "...examined using multivariate logistic regression models". Is it multivariate or multivariable? There is a major difference between the two methodologies.

Response

Thank you for your reminding. We corrected “multivariate” into “multivariable” through the revised manuscript. Please refer to Abstract (line 21), 2.3. Data analysis (line 181), and Results (line 219, 224, 227).

Comment 2

In the methods section the authors have not specified whether they have tested for effect modification. To this end, they should examine one of the independent variables used in the model as a potential exposure of interest and then identify the corresponding confounders of interest. This is a very important step in the analysis, since it can provide further information on the association under investigation.

Response

Thank you for your comment. In the revised manuscript we added the results bivariable logistic regression analysis for the comparison with the results of multivariable logistic regression analysis after adjusting the effects of demographics and sexual orientation. We also added the results of examining the moderating effects of demographics and sexual orientation on the associations of the perceived risk, emotional problems, and information sources with having a mpox vaccination.

Abstract

“…age did not moderate the associations of perceiving a higher risk of contracting mpox, receiving mpox information from health-care providers, and depression with having a mpox vaccination.” Please refer to line 26-27.

Methods

The moderating effects of demographics and sexual orientation on the associations of information sources and emotional problems were also examined.” Please refer to line181-183.

Results

The interactions of age with depression, perceived risk of contracting mpox, and receiving mpox information from healthcare providers further entered into a multivariable logistic regression analysis (Model II) to examine the moderating effects of age. However, the interactions were not significant associated with receiving a mpox vaccination (p > 0.05), indicating that age did not moderate the associations of depression, perceived risk of contracting mpox, and receiving mpox information from healthcare with receiving a mpox vaccination.” Please refer to line 217-223 and Table 3.

Comment 3

In the multivariable logistic regression model, I suggest testing also the 5-year increment of age, since it can reveal essential information otherwise not captured in the model.

Response

Thank you for your comment. Please refer to line 223-226 and Supplementary Table 1.

We also transformed age into 5-year increment of age and its association with receiving a mpox vaccination using multivariable logistic regression analysis (Supplementary Table 1). The result indicated that 5-year increment of age was still significantly associated with receiving a mpox vaccination (p < 0.001).

Comment 4

At the end of the discussion section the authors talk about the study limitations but do not reflect on the study strengths. What are this study's strengths? This information needs to be included as well.

Response

Thank you for your comment. We added the strength of this study into Discussion. Please refer to line 303-305.

The present study is the first one to examine the roles of sources of information and emotional problems on mpox in the uptake of vaccines in MSM. The results of this study provide evidence for developing intervention programs to enhance MSM’s intention to receive a mpox vaccination.

Round 2

Reviewer 3 Report

Comments and Suggestions for Authors

The manuscript “Factors related to monkeypox vaccine uptake among men who have sex with men in Taiwan: Roles of information sources and emotional problems” was improved with the revision done by authors. However, before the submission for publication the text should have minor revision:

1.       Change the name of the article to “Factors related to mpox vaccine uptake…”

2.       The numbers of references in the text should follow punctuation, and preferable be place at the end of sentence. Examples: Lines 50, 78, 93 to 104, and so on

3.       Write numbers under 10 as words and above as numerals. Examples: Lines 59 and 60 “… 75% for one dose and 80% for two doses…”; line 62 “…two-dose…”; lines 162 and 165 “…four-point scale…”; line 170 “…five-point scale…”; lines 224 and 226 “…five-year increment…”

4.       Avoid beginning a sentence with an abbreviation. Examples: Lines 148, 251, 267, and 280 ”Men who have sex with men…”

5.       Please correct to: Line 99 “…revealed that those ones…”; line 245 “…for this group of people”

6.       Lines 38 to 45. Please add a sentence according that “The clinical presentation of mpox cases with this outbreak has been atypical as compared with previously documented reports. Many cases in affected areas are not presenting with the classically described clinical picture for mpox, and few lesions, or no lesions, but anal pain and bleeding, genital and perineal/perianal alone lesions, lesions at different stages of development, and absence of prodromal period or constitutional symptoms appearing after lesions are observed.

Author Response

We appreciated your valuable comments. As discussed below, we have revised our manuscript with underlines based on your suggestions. Please let us know if we need to provide anything else regarding this revision.

Comment

  1. Change the name of the article to “Factors related to mpox vaccine uptake…”

Response

Thank you for your reminding. We changed “Monkeypox” into “Mpox” in the title. Please refer to line 2.

Comment

  1. The numbers of references in the text should follow punctuation, and preferable be place at the end of sentence. Examples: Lines 50, 78, 93 to 104, and so on

Response

We moved the numbers of references to the end of sentence thorough the manuscript.

Comment

  1. Write numbers under 10 as words and above as numerals. Examples: Lines 59 and 60 “… 75% for one dose and 80% for two doses…”; line 62 “…two-dose…”; lines 162 and 165 “…four-point scale…”; line 170 “…five-point scale…”; lines 224 and 226 “…five-year increment…”

Response

Thank you for your reminding. We revised them in the revised manuscript.

Comment

  1. Avoid beginning a sentence with an abbreviation. Examples: Lines 148, 251, 267, and 280. ”Men who have sex with men…”

Response

Thank you for your reminding. We revised them in the revised manuscript accordingly.

Comment

  1. Please correct to: Line 99 “…revealed that those ones…”; line 245 “…for this group of people”

Response

Thank you for your suggestion. We revised them accordingly.

Comment

  1. Lines 38 to 45. Please add a sentence according that “The clinical presentation of mpox cases with this outbreak has been atypical as compared with previously documented reports. Many cases in affected areas are not presenting with the classically described clinical picture for mpox, and few lesions, or no lesions, but anal pain and bleeding, genital and perineal/perianal alone lesions, lesions at different stages of development, and absence of prodromal period or constitutional symptoms appearing after lesions are observed.

Response

We appreciate your suggestion. We added the sentences into the revised manuscript. Please refer to line 45-50.